# Plasma Proteomic Analysis Identified Proteins Associated with Faulty Neutrophils Functionality in Decompensated Cirrhosis Patients with Sepsis

**DOI:** 10.3390/cells11111745

**Published:** 2022-05-25

**Authors:** Rashi Sehgal, Navkiran Kaur, Rakhi Maiwall, Gayatri Ramakrishna, Jaswinder Singh Maras, Nirupma Trehanpati

**Affiliations:** 1Department of Molecular and Cellular Medicine, Institute of Liver and Biliary Sciences, New Delhi 110070, India; rashi.sehgal19@gmail.com (R.S.); rgayatri@ilbs.in (G.R.); 2Amity Institute of Biotechnology, Amity University, Noida 201313, India; navkirank@amity.edu; 3Department of Hepatology, Institute of liver and biliary sciences, New Delhi 110070, India; rakhi_2011@yahoo.co.in

**Keywords:** neutrophils, monocytes, sepsis, NETosis, proteomic and autophagy

## Abstract

Decompensated cirrhosis (DC) is susceptible to infections and sepsis. Neutrophils and monocytes provide the first line of defense to encounter infection. We aimed to evaluate proteins related to neutrophils functionality in sepsis. 70 (DC), 40 with sepsis, 30 without (*w*/*o*) sepsis and 15 healthy controls (HC) plasma was analyzed for proteomic analysis, cytokine bead array, endotoxin, cell free DNA and whole blood cells were analyzed for nCD64-mHLADR index, neutrophils-monocytes, functionality and QRT-PCR. nCD64-mHLADR index was significantly increased (*p* < 0.0001) with decreased HLA-DR expression on total monocytes in sepsis (*p* = 0.045). Phagocytic activity of both neutrophils and monocytes were significantly decreased in sepsis (*p* = 0.002 and *p* = 0.0003). Sepsis plasma stimulated healthy neutrophils, showed significant increase in NETs (neutrophil extracellular traps) and cell free DNA (*p =* 0.049 and *p* = 0.04) compared to *w*/*o* sepsis and HC. Proteomic analysis revealed upregulated- DNAJC13, TMSB4X, GPI, GSTP1, PNP, ANPEP, COTL1, GCA, APOA1 and PGAM1 while downregulated- AHSG, DEFA1,SERPINA3, MPO, MMRN1and PROS1 proteins (FC > 1.5; *p* < 0.05) associated to neutrophil activation and autophagy in sepsis. Proteins such as DNAJC13, GPI, GSTP1, PNP, ANPEP, COTL1, PGAM1, PROS1, MPO, SERPINA3 and MMRN1 showed positive correlation with neutrophils activity and number, oxidative burst activity and clinical parameters such as MELD, MELD Na and Bilirubin. Proteomic analysis revealed that faulty functionality of neutrophils may be due to the autophagy proteins i.e., DNAJC13, AHSG, TMSB4X, PROS1 and SERPINA3, which can be used as therapeutic targets in decompensated cirrhosis patients with sepsis.

## 1. Introduction

Liver cirrhosis is the end stage of chronic liver disease due to viral (hepatitis B and hepatitis C) and non-viral etiologies (alcohol intake, non-alcoholic fatty liver) [1,2]. Neutrophils act as the first line of defense, followed by monocytes in host defense and inflammation. Cirrhotic showed impaired phagocytic ability, high resting oxidative burst activity of neutrophils, and decreased HLA-DR expression of monocytes [3,4,5], although peripheral blood monocytes showed no significant difference in TLR-4 expression between controls and cirrhotic groups [6]. This directly or indirectly influences the ability of T, B cells and adaptive immunity [7].

In cirrhosis, translocation of gut bacteria to the liver is enhanced due to increased membrane permeability, causing endotoxemia and systemic inflammation with dysregulated immune cell activation. Compromised immunity of cirrhotic patients made them more prone to multiple infections, including spontaneous bacterial peritonitis (SBP), urinary tract infection, pneumonia, cellulitis resultant sepsis and septic shock with multiple organ dysfunction in 20–40% patients [7,8,9].

Chemokine receptors CXCR1 and CXCR2 drive neutrophil migration to the inflammation site in response to its ligand CXCL8/IL-8, leading to the hepatic inflammatory response [10], although infectious and inflammatory conditions significantly extend the life span of short-lived neutrophils making them dysfunctional [11]. Dysfunctional neutrophils contribute to greater morbidity and mortality with the rise in the sepsis index, which is calculated by CD64 expression of neutrophils and HLA-DR expression of monocytes [12]. Additionally, degranulated neutrophils have the ability to secrete their DNA as neutrophil extracellular traps (NETs) decorated with histones and cytotoxic proteins, i.e., elastase, myeloperoxidase (MPO), cathepsin G, lactoferrin, pentraxin 3, gelatinase, proteinase 3 (PR3), LL37, and peptidoglycan. NETs are used to trap and neutralize pathogens by bactericidal activity [13]. However, a decrease in the antimicrobial capability of neutrophils occurred due to the deficient release of MPO in decompensated alcoholic cirrhosis [14]. Most of the studies have observed both monocytes and neutrophils functionality in cirrhotic patients but very few have identified proteins linked to monocyte activity. Different proteomic studies have identified proteins from neutrophilic cellular and subcellular fractions, including granules, secretory vesicles, and plasma membranes [15,16,17].

During sepsis, autophagy uses a cellular adaptive defensive method for limiting cell damage and apoptosis, as well as removing bacteria and pathogens. The autophagy process even showed altered effects on the different stages of neutrophil differentiation. Any deficiency in autophagy in neutrophils significantly reduced ROS generation, NADPH oxidase, and degranulation in vitro and in vivo. Both autophagy and ROS generation are mainly required for chromatin decondensation in NETosis, and impaired autophagy causes a decrease in NETosis, leading to the survival of neutrophils which directly increase the mortality in sepsis patients [18,19,20].

Our aim was to identify the unique proteins related to neutrophils functionality in decompensated cirrhosis with sepsis.

## 2. Materials and Methods

### 2.1. Study Patients and Blood Sampling

We enrolled 70 decompensated cirrhosis patients: without Sepsis (*n =* 30), with sepsis (*n =* 40) and healthy control (HC, *n =* 15) at Institute of Liver and Biliary Sciences (ILBS), New Delhi between 2017 and 2019. This study was approved by the Research and Institutional ethics committee of ILBS with IEC No IEC/2016/45/NA/C2, and informed consent was obtained from all the subjects enrolled in the study. In this longitudinal study, all the patients were closely monitored from the time of admission and studied at baseline, day 3 and 7. This study was carried out in accordance with the ethical standards of the Helsinki declaration.

Patients with history of any hepatitis infection (HBV, HCV etc.), with HCC or any other site malignancy or any other co-morbidities, and no consent given were excluded from the study. The clinical as well as biochemical assessment of all the enrolled patients with or without SIRS (systemic inflammatory response syndrome) or sepsis patients was done according to the treating physician’s direction. Patients were recruited based on the following criteria:

Decompensated cirrhosis without (*w*/*o*) sepsis was diagnosed when there was no evidence of SIRS, or infection on any of the cultures and ultrasonography.

Decompensated cirrhosis with sepsis were diagnosed based on the presence of SIRS, SOFA (sequential organ failure assessment) score with an infection, confirmed either by cultures or by imaging.

Four criteria for the diagnosis of SIRS were used [21]:(a)Temperature >38.0 °C or <36.0 °C(b)TLC (Total leukocyte count) < 4 or >12 × 10^9^/L(c)Pulse > 90 beats/minutes(d)RR (Respiratory rate) >20 or <32 beats/minutes

According to the institutional protocol, medications were given to:

**DC patients without sepsis:** For standard indications (i.e., AKI or refractory ascites), IV fluids and albumin are given however prophylactics and third generation cephalosporin antibiotics were given to the patients admitted in ICU/HDU.

**DC patients with Sepsis:** Broad-spectrum antibiotics (which covers gram −ve, gram +ve, and fungal infections) were given to the patients according to the site (lungs, ascites, skin, urine etc.), severity (sepsis with organ dysfunction or septic shock), and acquisition of the infection (community vs. healthcare or nosocomial infections).

### 2.2. Blood Sampling

10–15 mL of peripheral blood was collected at room temperature in an ethylene diamine tetraacetic acid (EDTA) coated and heparin tubes from patients and healthy subjects at the time of recruitment in the study. Peripheral blood was used for whole blood immunophenotyping, phagocytic activity, oxidative burst activity, expression of neutrophilic CD64 and monocytic HLA-DR (nCD64-mHLADR Index), and neutrophils isolation. From the rest blood, plasma was separated for assessment of cytokines, endotoxin, and other circulating proteins, whereas plasma devoid blood was used for isolation peripheral blood mononuclear cells (PBMCs).

### 2.3. Endotoxin Estimation

Endotoxin levels were estimated using PierceTM Chromogenic Endotoxin Quant Kit (Thermo fisher Scientific, #A39552). All reagents and standards were prepared as per the manufacturer protocol freshly before use. Plate was pre-equilibrated at 37 ± 1 °C heating block and maintained for whole procedure time. 50 µL of blanks, standards, and samples added/well followed by addition of 50 µL of the reconstituted amebocyte lysate reagent/well. After addition of amebocyte lysate reagent, the plate was removed from the heater and mixed gently by tapping 10 times on the sides of the plate and returned to the plate heater at 37 ± 1 °C. Chromogenic substrate was reconstituted with endotoxin free water and pre-warm at 37 ± 1 °C for 5 min before use. 100 µL of chromogenic substrate was added and mixed gently by tapping 10 times on the sides of the plate. 50 µL of stop solution (25% acetic acid) was added/well while keeping the plate at heater at 37 ± 1 °C and mixed gently by tapping 10 times. Optical density (OD) of the plate was immediately taken at 405 nm using Multiskan EX (Thermo fisher Scientific, Waltham, MA, USA #51118170). Average absorbance of blank to be subtracted from all standards and samples replicates for calculation of mean ∆ absorbance. Standard curve was prepared using blank subtracted absorbances for the standards with endotoxin concentration in EU/mL. Linear regression with the coefficient of determination was used to determine the endotoxin concentration of each sample.

### 2.4. Multi-Parametric Whole Blood Immunophenotyping

Neutrophils and monocytes were characterized in whole blood using specific antibodies against surface markers labelled with different fluorochromes. Neutrophils were characterized using anti-CD11b at 1:100 dilution (Clone ICRF44, PecY7, Biolegend 301322), anti-CD66b at 1:40 dilution (Clone G10F5, PE, Biolegend 305106, San Diego, CA, USA), anti-CXCR2 at 1:40 dilution (Clone 5E8/CXCR2, FITC, Biolegend 320704), and anti-CXCR1 at 1:40 dilution (Clone 5A12, BV510, BD Biosciences 743420). Monocytes were characterized using anti-CD14 at 1:40 dilution (Clone M5E2, FITC, BD Biosciences 555397), anti-CD16 at 1:30 dilution (Clone B73.1, BV450, BD Biosciences 561310), and anti-HLA-DR at 1:40 dilution (Clone L243, BV510, Biolegend 307646) (BD Biosciences, San Jose, CA, USA and Biolegend San Diego, CA, USA).

Briefly, 100 µL of whole blood was incubated with appropriate concentrations of surface antibodies for 30 min at room temperature in the dark. After that, freshly prepared 1× RBC lysis solution added in the cells and incubated for 5 min at room temperature, dark and washed with 1× PBS. A minimum of 100,000 events were acquired using BD FACS VERSE, and all relevant data was analyzed by using FlowJo software (version 10).

### 2.5. Phagocytic Activity

Phagocytic activity was performed using manufacturer protocol (Celonic, Basel, Switzerland #341060). All the reagents i.e., opsonized FITC labelled *E. coli* bacteria, quenching solution, lysis solution and wash solution provided in the ready to use phagotest kit from Celonic, Switzerland #341060. 100 µL of heparin blood was used and mixed with 20 µL precooled *E. coli* bacteria (4 × 10^7^ bacteria). Both test sample and control tube were incubated in water bath for 10 min at 37 °C. After 10 min, tubes were kept on ice to stop phagocytosis. 100 µL of ice-cold quenching solution added to each tube and mixed properly using vortex mixer. 3 mL of wash solution added twice to the tubes and centrifuged for 5 min, 250× *g*, 2–8 °C and supernatant was discarded. After that, pre-warmed (room temperature) 1× lysis solution was added to the samples, vortexed and incubated for 20 min at room temperature, then centrifuged for 5 min, 250× *g*, 2–8 °C and supernatant was discarded. Again, 3 mL of wash solution added to the tubes and centrifuged for 5 min, 250× *g*, 2–8 °C. After washing twice, samples were ready to be acquired within 60 min. 100,000 events/tube acquired in BD FACS VERSE. Neutrophils and monocytes were gated to analyze the phagocytic activity using FlowJo (version 10).

### 2.6. Oxidative Burst Activity 

Oxidative burst activity was performed using manufacturer protocol (Celonic, Basel, Switzerland #341058). All the reagents i.e., opsonized *E. coli* bacteria (particulate stimulus), FMLP (low physiological stimulus), PMA (high stimulus), dihydrorhodamine 123, lysis solution and wash solution provided in the ready to use oxidative burst kit from Celonic, Switzerland #341058. Accordingly, 100 µL of heparin blood was incubated with 20 µL of each with final concentration; precooled *E. coli* bacteria (2 × 10^7^ bacteria), FMLP (5 µM), and PMA (8.1 µM), and incubated for 10 min at 37 °C in pre-warmed water bath. Both FMLP and PMA reagents were freshly prepared in wash solution immediately before the experiment. 

After incubation, 20 µL of dihydrorhodamine 123 (substrate solution) was added in all tubes, except unstained and incubated for 10 min at 37 °C in water bath, followed by addition of the 1ml of pre warmed 1× lysis solution. Tubes were vortexed and incubated further for 20 min at room temperature. Again, 3 mL of wash solution added to the tubes, centrifuged for 5 min at 250× *g* at 2–8 °C, and supernatant was discarded. After washing twice, samples were ready to be acquired within 30 min. 100,000 events/tube were acquired in BD FACS VERSE. Neutrophils and monocytes were gated to analyze the oxidative burst activity using FlowJo (version 10).

### 2.7. Expression of Neutrophilic CD64 and Monocytic HLA-DR (nCD64-mHLADR Index)

The expression of nCD64 and mHLA-DR was used for calculating nCD64-mHLADRindex by using BD Quantibrite™ CD64/CD45 (BD Biosciences, San Jose, CA, USA #340768) and BD Quantibrite™ Anti-HLA-DR/Anti-Monocyte (BD Biosciences, San Jose, CA, USA #340827), as per the protocol. Briefly, for staining, 50 μL of whole blood was mixed with 20 μL of conjugated monoclonal antibody and incubated for 60 min in the dark at 20–25 °C, room temperature. After incubation, 2 mL of 1× BD FACS lysing solution (BD Biosciences, San Jose, CA, USA #349202) was added and further incubated for 30 min. Finally, cells were washed and quantitation acquisition was done in BD FACS Calibur using BD CellQuest™ software.

nCD64-mHLADR index was calculated following this formula:nCD64-mHLADR Index *=* nCD64 × 100/mHLA-DR
where nCD64 *=* Geometric mean of CD64 neutrophils.

mHLA-DR *=* Geometric mean of HLA-DR monocytes.

### 2.8. Isolation of Neutrophils

Neutrophils were isolated using polymorphoprep (AXIS SHEILD, Oslo, Norway; #AXS-1114683) at room temperature immediately after collection of peripheral blood. Anti-coagulated whole blood was layered on top of polymorphoprep layer in 1:1 ratio in a 15 mL centrifuge tube and centrifuged at 500× *g* for 30–35 min at room temperature. After centrifugation, two leucocytes’ bands were visible. The top band at the sample interface consisted of mononuclear cells and the lower band of neutrophils. The lower band of neutrophils harvested into a new centrifuge tube with pasteur pipette. Cells were washed with 1× PBS and centrifuged at 400× *g* for 10 min. Supernatant removed and resuspended the cells in 1 mL of PBS. Cells counted using hemocytometer. Purity of the cell preparation can be determined by flow cytometry and viability by trypan blue exclusion. Isolated neutrophil cells were further used for NETosis.

### 2.9. NETosis

Round coverslips were coated with 1x poly-lysine (0.01%, Sigma Aldrich, St. Louis, MO, USA # P8920), placed in 12 well plate and incubated in CO_2_ incubator overnight. After that, the plate was removed from CO_2_ incubator, washed with autoclaved distilled water, and dried. 5 × 10^5^ isolated neutrophil cells plated/well in RPMI media (GIBCO, Thermofisher Scientific, Waltham, MA, USA #22400-089) with 10% FBS (Fetal bovine serum, GIBCO, Thermofisher Scientific, MA, USA #10270-106) and 1% penicillin-streptomycin (HIMEDIA, Mumbai, Maharashtra, INDIA, #A001) and incubated in CO_2_ incubator for 2 h for the attachment of neutrophils. After 2 h, all media were removed and replaced with fresh RPMI media along with stimulations i.e., PMA (50 ng/uL, Thermofischer Scientific, MA, USA #16561-29-8), LPS (100 ng/uL, Elabsciences, Houston, TX, USA #E-BC-R188), rIL-8 (1000 pg/uL, Peprotech #200-08) and PLASMA (1:10 dilution). After that, the plate was incubated again in CO_2_ incubator for 4–6 h. After that, media was removed from all wells and collected for estimation of cell free DNA, cells fixed with 4% paraformaldehyde for 10–15 min. After 15 min, cells were washed with 1× PBS and coverslips mounted on slides with DAPI-mount (Vectashield, Burlingame, CA, USA #H-1200). Slides were used to take images of NETs using a Nikon Intense light Microscope.

### 2.10. Estimation of Cell Free DNA

For cell free DNA estimation, media removed after 4–6 h of stimulation was used. Salmon sperm DNA (10 mg/mL, Thermofischer Scientific, MA, USA #15632011) was used as standards at different concentrations. Samples and standards were stained with sytox green (5 mM, Thermofischer Scientific, MA, USA, #S7020) for 5 min and plated in duplicates in black cell culture plates (SPL life sciences, Gyeonggi-do, Korea, #33196). Fluorescence of sytox green was detected at 485/520 (Exi/Emi) using FLUOstar Optima (BMG LABTECH, Ortenberg, Germany).

### 2.11. Cytokine Multiplex Bead Array Assay

To understand the significance of various cytokines and growth factors associated with sepsis and neutrophils, we analyzed forty-one cytokines, chemokines, and growth factors such as IL-1RA, IL-1β, IL-2, IL-12 (p40), IL-27, IL-18, IL-10, IL-17A, IFN-γ, TNF-α, IL-4, IL-6, IL-33, TGF-β1, IL-8, MIF, MIP-1α, MIP-1β, MIP-3α, ITAC, MCP-1, FRACTALKINE, ENA78, IP-10, EOTAXIN, Angiopoientin, MCSF, G-CSF/CSF3, GM-CSF, HGF, LEPTIN, TPO, VEGF-A, MMP7, MMP8, MMP1, MMP12, MMP13, P SELECTIN, E SELECTIN and TREM1 in plasma by using custom multiplex PROCATAPLEX (THERMOFISCHER, USA), following the complete details on Luminex xponent 3.1TM Rev.2 (Luminex Corporation, 12212 Technology Boulevard, Austin, TX, USA). Standard curve was drawn using standards provided in the kit. Values in samples were determined corresponding to the standard curve drawn. The lower limits of detection of the test for each analyte is provided in the Appendix A.

### 2.12. Plasma Proteomic Analysis

In brief, plasma samples used for proteomic analysis were diluted with 0.1% formic acid in 1:5 ratio and run through an AKTA pure protein purification system (Cytiva) using UNICORN 7.1 software in order to deplete albumin. Albumin depleted supernatant was collected and used for protein estimation using Bradford reagent (Cepham life sciences, Fulton, MD, USA #10478). A total of 30 ug of proteins were dissolved in 50 mM ammonium bicarbonate and reduced by dithiothreitol (DTT) for 40 min at 56 °C. Proteins were then alkylated by 20 mM iodoacetamide (IAA) for 40 min at room temperature in the dark. The 30 μg of protein samples were then kept for digestion with 1 μg sequencing grade trypsin (Promega, Madison, WI, USA #V5111) at 37 °C. The reaction was stopped by acidifying the peptides using 0.1% formic acid followed by desalted on reversed-phase C-18 spin columns (Thermofisher scientific, #89870). The peptides were then eluted using 50 µL of 60% acetonitrile in 0.1% formic acid and concentrated in a speed vac machine. The peptides were finally acidified with 2% acetonitrile, 0.1% trifluoroacetic acid in 0.1% formic acid. These peptides were subjected to nano-electrospray ionization and Tandem mass spectrometry (MS/MS) using Q-ExactiveTM Plus (Thermo Fisher Scientific, San Jose, CA, USA) at the collision-induced dissociation mode with the electrospray voltage was 2.3 kV. Further data was analyzed using proteome discoverer (version 2.0, Thermo Fisher Scientific, Waltham, MA, USA). Uniprot Homo sapiens (Human) database (UP000005640) with Mascot algorithm (Mascot 2.4, Matrix Science, Boston, MA, USA) was used. Identified proteins were subjected to standard statistical analysis and network and pathway analysis [22].

### 2.13. Quantitative RT-PCR Analysis

Total RNA was isolated from neutrophils using miRVANA kit (Thermofischer Scientific, Waltham, MA, USA, #AM1560). cDNA was prepared from RNA using High-Capacity cDNA Kit (ABI, Thermo Fisher Scientific, Waltham, MA, USA, #4368813). mRNA expression of genes like DNAJC13, TMSB4X, AHSG, ATG5, ATG12, HMGB1, and HIF-1α were analyzed. The primer sequences of the genes are listed in Appendix A. 18s RNA served as endogenous control for normalization. Relative expression was analyzed using SYBR Green (Thermofischer Scientific, Waltham, MA, USA, #A25742) in a VIIA7 real-time PCR machine (ABI, Whitefield, Bangalore, India).

### 2.14. Statistical Analysis

Data were analyzed using the statistical software Prism (version 6; GraphPad Software, San Diego, CA, USA) and SPSS version 22 (IBM Corp Ltd., Armonk, NY, USA). The comparison for continuous data is carried by using One-way ANOVA/Kruskal–Wallis test followed by probability adjustment by the Mann–Whitney test or by Bonferroni test post-hoc comparison as appropriate and it is represented as mean ± standard deviation (SD). The *p*-values <0.05 were considered statistically significant.

## 3. Results

Baseline characteristics of 70 DC patients, 30 patients *w*/*o* sepsis (age 47 ± 5 years, 87% males) and 40 patients with sepsis (43 ± 9, 97% males) and 15 age-matched healthy controls were analyzed at the time of admission and enrolment in the study. Alcohol was the predominant etiology (70%) in DC patients. Sepsis patients showed a significant increase in total bilirubin, AST levels, INR, PCT, lactate, MELD Na, SOFA score, creatinine, TLC, and pulse compared to *w*/*o* sepsis in Table 1.

Plasma endotoxin levels were found to have increased in the sepsis patients compared to *w*/*o* sepsis and HC (*p* = 0.028 and *p* = 0.042) (Figure 1A).

### 3.1. Neutrophils in Sepsis

#### 3.1.1. Increase in Neutrophils and Neutrophil-to-Monocyte Ratio in Sepsis Patients

Differential leukocyte count (DLC) showed an increase in the percentage of neutrophils and neutrophil-to-monocyte ratio (NMR) in sepsis patients (*p =* 0.0076, *p =* 0.008) compared to *w*/*o* sepsis. No significant difference was observed in the percentage of monocytes between the groups (Figure 1B).

#### 3.1.2. Assessment of Neutrophilic CD64 and Monocytic HLADR Expression in Sepsis

Based on the expression of neutrophils and monocytes in differential leukocyte count, we have checked the expression of nCD64 and mHLA-DR in the peripheral blood of the study groups using BD Quantibrite. nCD64-mHLADR index was found significantly increased in sepsis and *w*/*o* sepsis patients compared to HC (*p* < 0.0001 and *p =* 0.004), although it was two-fold more increased in sepsis patients compared to *w*/*o* sepsis, but no significant difference was observed between sepsis and *w*/*o* sepsis (Figure 1C). Therefore, this index suggests that neutrophils can play a crucial role in the identification of sepsis at an initial stage.

#### 3.1.3. Decreased Neutrophils Chemoattractant Receptors in Sepsis Patients

High dimensional flow cytometry analysis in peripheral blood also revealed that the percent neutrophil was significantly increased in sepsis patients compared to *w*/*o* sepsis and HC (*p* < 0.0001 and *p =* 0.008) (Figure 1D,E) and supported differential leukocyte count results. Further, the percent expression of CD11b^+^CD66b^+^ on neutrophils shows no difference but chemotactic cytokine receptor CXCR1 and CXCR2 on CD11b^+^CD66b^+^ were found to decrease in sepsis patients compared to *w*/*o* sepsis and HC (Figure 1E). It clearly suggested that neutrophils are increased in sepsis, but they are not activated and had low chemotactic activity.

No difference in percentages of neutrophils was observed at follow-up time points in *w*/*o* and with sepsis patients. However, percentage frequencies of CD11b^+^CD66b^+^ and CXCR2^+^ on CD11b^+^CD66b^+^ were decreased in sepsis patients at day 7 (*p =* 0.005 and *p =* 0.047) compared to day 0. However, percentage frequencies of CD11b^+^CD66b^+^, CXCR1^+^ and CXCR2^+^ on CD11b^+^CD66b^+^ were increased in *w*/*o* sepsis patients at day 7 (*p =* 0.007, *p =* 0. 018, *p =* 0.008) compared to day 0 (Appendix A).

Based on etiologies in decompensated cirrhosis patients, i.e., alcohol (ALC) and other etiology (NASH and cryptogenic), we have found no difference in the percentage of neutrophils CD11b^+^CD66b^+^, CXCR1^+^, and CXCR2^+^ on CD11b^+^CD66b^+^ cells in decompensated cirrhosis patients *w*/*o* and with sepsis (Appendix A). Considering AKI (acute kidney injury) as a decompensating besides sepsis, we found a decrease in the percentage of neutrophils, CD11b^+^CD66b^+^, CXCR1^+^, and CXCR2^+^ on CD11b^+^CD66b^+^ cells in AKI patients compared to no AKI in decompensated cirrhosis with sepsis (Appendix A).

All sepsis patients received broad-spectrum antibiotics leading to the resolution of the infection in a few, contrasting with others who were not able to resolve the infection. There was no difference in percent expression of total neutrophils, percent expression of CD11b^+^CD66b^+^ neutrophils, CXCR1^+^, and CXCR2^+^ on CD11b^+^CD66b^+^ cells in sepsis patients (Appendix A).

#### 3.1.4. Decreased Phagocytic Activity of Neutrophils in Sepsis

We have analyzed the functionality of neutrophils by measuring phagocytosis and oxidative burst activity in the patient groups. Neutrophils showed a significant decrease in phagocytic activity in DC patients with and *w*/*o* sepsis compared to HC (*p =* 0.002 and *p =* 0.02) (Figure 2A,B).

Oxidative burst activity of neutrophils was observed spontaneously and with different stimulus, i.e., FMLP (low physiological stimulus), PMA (high stimulus), and *E. coli* (particulate stimulus). Spontaneous (SPT) and FMLP stimulated oxidative burst activity of neutrophils showed no difference between the groups. PMA, being a strong stimulant, still showed decreased oxidative burst activity of neutrophils in sepsis compared to *w*/*o* sepsis and HC. However, *E. coli* stimulated neutrophils showed a significant increase in oxidative burst compared to spontaneous in all the groups, i.e., HC (*p* < 0.0001), *w*/*o* sepsis (*p* < 0.0001) and sepsis (*p* < 0.0001), but *E. coli* stimulated neutrophils had decreased oxidative burst activity of neutrophils in sepsis patients (*p =* 0.028) compared to HC (Figure 2C,D). This clearly suggests sepsis patients had an increase in neutrophil numbers merely with the fault in functionality, i.e., phagocytosis and oxidative burst activity.

No significant difference in phagocytic and oxidative burst activity of neutrophils was observed in different time points between the groups (Appendix A).

#### 3.1.5. Increase in Neutrophil Extracellular Traps and Cell Free DNA Concentration in Sepsis Plasma

NETs are extracellular web-like DNA arrangements decorated with various antimicrobial proteins. Healthy neutrophils were stimulated with LPS, PMA, rIL-8 as positive controls, and the patient’s plasma for assessing NETs formation as well as the release of cell-free DNA in the supernatant. Plasma from sepsis patients have more NETs formation compared to *w*/*o* sepsis and HC plasma. As positive controls, rIL-8 stimulated neutrophils showed many NETs and LPS stimulated neutrophils showed very few NETs (Figure 2E). Further, cell-free DNA was also found to be significantly increased with sepsis plasma compared to *w*/*o* sepsis (*p =* 0.049) and HC plasma (*p =* 0.04). Though *w*/*o* sepsis patient plasma also showed a significant increase in cell-free DNA compared to HC (*p =* 0.044) (Figure 2F).

No significant difference in cell free DNA concentration was observed at different time points between the groups (Appendix A).

### 3.2. Monocytes in Sepsis

#### 3.2.1. Reduced mHLA-DR in Sepsis Patients

Expression of HLA-DR on monocytes (mHLA-DR) in the peripheral blood was significantly reduced in sepsis patients (*p =* 0.045) compared to HC, but no significant difference was observed between sepsis and *w*/*o* sepsis patients (Figure 3A,B). Anti-inflammatory monocytes (non-classical monocytes) were found to be decreased in sepsis patients (*p =* 0.007) compared to *w*/*o* sepsis. Whereas classical and intermediate monocytes showed no difference in the groups (Figure 3C). Expression of HLA-DR on non-classical monocytes were found to have significantly decreased in sepsis patients compared to *w*/*o* sepsis and HC (*p =* 0.042 and *p =* 0.03) but no difference in expression of HLA-DR on classical and intermediate monocytes (Figure 3D).

There was no difference in the expression of mHLA-DR, classical and intermediate monocytes at follow-up time points in sepsis and *w*/*o* sepsis patients. The percentage frequencies of non-classical monocytes were decreased in sepsis patients at day 7 (*p =* 0.047) compared to day 0. There was no difference in the expression of HLA-DR at follow-up time points in classical and intermediate monocytes, but HLA-DR expression on non-classical monocytes was found significantly increased in *w*/*o* sepsis patients at day 7 (*p =* 0.01) compared to day 0 (Appendix A).

We found an increase in HLA-DR on monocytes at day 7 in the resolution of sepsis group compared to day 0, but no resolution group showed a decrease in HLA-DR on monocytes at day 7 compared to day 0. Classical, intermediate, and non-classical monocytes also showed no difference (Appendix A).

#### 3.2.2. Decrease in Phagocytic and Oxidative Burst Activity of Monocytes in Sepsis

Phagocytosis and oxidative burst activities are hallmarks of functional monocytes. However, both the activities were significantly reduced in sepsis and *w*/*o* sepsis patients (*p =* 0.0003, and *p =* 0.003) compared to HC, but no significant difference was observed between sepsis and *w*/*o* sepsis (Figure 4A,B).

Spontaneous (SPT) and FMLP stimulated monocytes displayed no difference in oxidative burst activity in all groups. PMA stimulated monocytes also showed a significant decrease in oxidative burst activity in sepsis compared to HC (*p =* 0.03). However, *E. coli* stimulated monocytes had increased oxidative burst activity compared to spontaneous activity in HC (*p =* 0.007), *w*/*o* sepsis (*p =* 0.002) and sepsis (*p =* 0.03) (Figure 4C,D).

No significant difference in both the phagocytic and oxidative burst activity of monocytes was observed at different time points between the groups (Appendix A).

### 3.3. Relation of Soluble Factors in Sepsis

Plasma samples were analyzed for soluble factors using proteomics, as well as cytokines, chemokines, and growth factors using cytokine bead array (Figure 5A).

#### 3.3.1. Neutrophils Specific Plasma Proteins in Sepsis

Principal component analysis (PCA) revealed a clear separation of plasma proteins in three different groups (Figure 5B). Further, the Venn diagram showed all the differentially expressed upregulated and downregulated proteins between the groups. A total of 30 upregulated proteins were uniquely expressed in sepsis and HC, 34 in *w*/*o* sepsis and HC, and 26 in sepsis and *w*/*o* sepsis, while 12 downregulated proteins were uniquely expressed in sepsis and HC, 32 in sepsis and HC, and 24 in sepsis and *w*/*o* sepsis (Figure 5C).

Gene ontology (GO) annotation analysis explained these uniquely expressed proteins, in their biological significance, cellular localization, and molecular functions. It was observed that upregulated proteins in sepsis patients compared to HC were DNAJC13, GPI, GSTP1, PNP, ANPEP, COTL1, GCA, PGAM1, and TMSB4X and compared to *w*/*o* sepsis patients were GPI, DNAJC13, and APOA1 (Neutrophil degranulation, neutrophil activation, neutrophil-mediated immunity, and secretory granule lumen (GO:0043312, GO:0002283, GO:0002446, and GO:0034774)) (Figure 5D).

GO analysis showed that most of the downregulated proteins were not related to neutrophils activity in sepsis patients. However, few downregulated proteins in sepsis patients compared to HC were AHSG, and PROS1, whereas compared to *w*/*o* sepsis, patients were AHSG, PROS1, DEFA1, SERPINA3, MPO, and MMRN1 (Azurophil granule, azurophil granule lumen and secretory granule lumen (GO:0042582, GO:0035578 and GO:0034774)) (Figure 5D). Out of these upregulated and downregulated proteins, some proteins, i.e., DNAJC13, AHSG, TMSB4X, PROS1, and SERPINA3 were known to induce the autophagy process, which helps in the neutrophil differentiation and its functionality [20]. Heatmap clearly shows the difference in these upregulated and downregulated proteins between all the groups (Figure 5E).

#### 3.3.2. Increased Cytokines in Sepsis

A total panel of 41 cytokines, chemokines, and growth factors associated with sepsis were used in the cytokine bead array assay. We found an increase in the expression of IL-6, IL-8, IL-10, IL-18, IL-33, IP-10, MIF, MIP1a, MIP1β, MIP3a, LEPTIN, GCSF, MCSF, and ESelectin in sepsis plasma compared to *w*/*o* sepsis and HC (Figure 5F).

#### 3.3.3. Decrease Cytokines in Sepsis

Apart from a few cytokines which were significantly increased in sepsis patients, many were significantly decreased, including IL-1RA, IL-1β, IL-2, IL-4, IL-12p40, IL-17A, IL-27, TNF-α, TGF- β1, MCP-1, ENA78, Angiopoietin, TREM1, EOTAXIN, GM-CSF, HGF, PSELECTIN, MMP1, MMP8, MMP12, and MMP13 in sepsis and *w*/*o* sepsis patients compared to HC (Figure 5F). Some of these cytokines show contradictory results in sepsis patients when compared to published studies.

#### 3.3.4. Correlation of Neutrophils Functionality with Plasma Proteins and Cytokines in Sepsis

We have correlated the expression of neutrophils and their functionality with the biochemical parameters, all cytokines and proteins linked to neutrophils in sepsis patients. In sepsis patients, MELD and MELD Na scores were increased and we observed that neutrophil related proteins i.e., DNAJC13, GPI, GSTP1, PNP, ANPEP, COTL1, PGAM1, APOA1, PROS1, SERPINA3, MPO, and MMRN1 were positively correlated with neutrophils, oxidative burst activity of neutrophils, MELD, MELD Na and bilirubin.

In addition to the proteins, many cytokines, i.e., IL-1β, IL-2, IL-4, IL-12p40, IL-17A, IL-18, IL-27, IL-33, IFN-γ, TNF-α, TGF-β1, MCP1, MIP1β, MIP3a, MMP1, MMP7, HGF, ENA78, TREM1, VEGF-A, MIF, EOTAXIN, PSELECTIN, and IP-10 were found to be positively correlated with neutrophils and their oxidative burst activity. Only IL-10 and MCSF were also found to be positively correlated with CD11b^+^CXCR1^+^ neutrophils. On the other hand, IL-2, IL-17A, IL-27, IL-33, MIP1a, MIP1β, MMP7, TPO, ESELECTIN, MIF, and EOTAXIN were found negatively correlated with the phagocytic activity of neutrophils.

Apart from neutrophils and their functionality, upregulated proteins, i.e., DNAJC13, GPI, GSTP1, PNP, ANPEP, COTL1, GCA and PGAM1 were also found positively correlated with many cytokines and growth factors i.e., IL-1β, IL-2, IL-4, IL-8, IL-12p40, IL-17A, IL-18, IL-27, MIP-1β, MIP3a, MMP1, MMP7, MMP8, IFN-γ, TNF-α, TGF-β1, MCP1, HGF, ENA78, TREM1, Angiopoietin, M-CSF, VEGF-A, ESELECTIN, PSELECTIN, MIF, EOTAXIN, and IP-10. Downregulated proteins, i.e., AHSG, PROS1, SERPINA3, MPO and MMRN1 were positively correlated and APOA1 was negatively correlated with IL-1β, IL-2, IL-4, IL-12p40, IL-17A, IL-18, IL-27, MIP1a, MIP-1β, MIP3a, IFN-γ, TNF-α, TGF-β1, MMP1, MMP7, MCP1, Angiopoietin, MCSF, HGF, ENA78, TREM1, VEGF-A, ESELECTIN, PSELECTIN, MIF, EOTAXIN, and IP-10 (Figure 5G and Appendix A).

#### 3.3.5. Differential Expression of Proteins and Cytokines at Follow-Up Time Points in Sepsis Patients

Using heatmaps and line diagrams, already found upregulated and downregulated proteins in sepsis patients were observed at different follow-up time points in sepsis patients. We found that the upregulated proteins, i.e., GPI, GCA, GSTP1, COTL1, PNP, TMSB4X, ANPEP and DNAJC13 were found to decrease at day 3 and day 7 compared to day 0, whereas PGAM1 and APOA1 increased at day3 later decreased at day 7 compared to day 0. Overall, the Eigen value (i.e., average of all values) of upregulated proteins shows decreased at day 7 compared to day 0 (Figure 6A). While downregulated proteins, i.e., AHSG, PROS1, SERPINA3, DEFA1, MPO, and MMRN1 were found increased at day 3 and day 7 compared to day 0. The Eigen value of downregulated proteins shows an increase at day 7 compared to day 0 (Figure 6B).

Using a heatmap, already found increased cytokines in sepsis patients at baseline were observed at different follow-up time points. We found that in sepsis patients at follow-up of day 3 and day 7, cytokines, i.e., IL-6, IL-8, IL-18, IP-10, MIF, and ESelectin were decreased and IL-10, IL-33, MIP1a, MIP1b, MIP3a, and MCSF were increased compared to day 0 (Figure 6C). We observed the same cytokines in *w*/*o* sepsis patients at follow-up of day 3 and day 7 time points, cytokines, i.e., IL-6, IL-8, IL-18, IL-33, MIF, MIP1a, MIP1b, and MIP3a were increased, and IP-10, MCSF, and ESelectin were decreased compared to day 0 (Figure 6D).

#### 3.3.6. QRT-PCR Analysis for Validation

QRT-PCR was done to validate DNAJC13, TMSB4X, and AHSG in the neutrophils. Expression of TMSB4X was found to be significantly increased in sepsis (*p =* 0.038) compared to HC and DNAJC13 and AHSG were found to decrease in sepsis compared to *w*/*o* sepsis (Figure 6E).

Apart from these proteins, expression of ATG12 was found to be increased in sepsis (*p =* 0.047), though another autophagy gene ATG5 was increased but not significant in sepsis. Further HIF-1α expression (*p =* 0.046) was found to decrease in sepsis; however, HMGB1 was decreased but not significant in sepsis compared to *w*/*o* sepsis patients (Figure 6F).

## 4. Discussion

Our study observed that neutrophils were increased in sepsis patients with decreased phagocytic and oxidative burst activity, while no difference in monocytes was observed with an expression of HLA-DR, phagocytic activity and oxidative burst activity of monocytes were decreased in sepsis patients. The nCD64-mHLADR index was found to be significantly increased in sepsis patients. NETs formation and cell-free DNA were found to be increased in sepsis patient plasma supernatants, with the rise in IL-8, IL-10, IL-18, etc., and a decline in IL-1β, IL-2, IL-4, etc. cytokines and DNAJC13, GPI, GSTP1, PNP, ANPEP, COTL1, GCA, PGAM1, TBMBS4X, and APOA1 proteins which play important roles in neutrophil activation and its immune response.

Sepsis is a life-threatening complication of an infection that progresses to septic shock induced by a dysregulated host response, resulting in a longer stay in hospitals with a higher mortality rate [23]. Sepsis patients showed downregulation of monocyte HLA-DR in bone marrow and blood, which correlates with their poor prognosis [24,25] and impaired neutrophil function, causing defects in innate immune response [7]. It was generally observed that sepsis patients develop leukocytosis somewhat due to the increase in neutrophils [26]. This correlates well with our study, as we found a significant decrease in monocyte HLA-DR and an increase in neutrophils with damaged neutrophil functionality.

Endotoxin levels were found to have increased in sepsis patients, which can prove helpful in understanding the decreased functionality of neutrophils and monocytes in sepsis. The interaction of endotoxin with neutrophils causes the alteration in oxidative, as well as the microbicidal properties of the neutrophils [27]. The membrane-bound form of CD14 (an important receptor for LPS) was found to be reduced in the presence of endotoxin, which causes a reduction in the stimulatory effect of LPS on monocytes [28]. A reduction in endotoxin levels in sepsis patients at day 7 shows the reduction in CD11b^+^CD66b^+^ neutrophils and non-classical monocytes compared to day 0. Resolution of sepsis shows a decrease in neutrophils and an increase in HLA-DR on monocytes at day 7 compared to day 0. The presence of AKI shows a decrease in neutrophils and CD11b^+^CD66b^+^ cells which clearly suggests neutrophil dysfunction due to AKI in sepsis patients. Many studies have developed scores to predict sepsis patient outcomes. Some of the scores used were APACHE and SOFA. Several studies suggested neutrophils as a successful marker for predicting sepsis [29,30]. Recent studies even suggested the DNI (delta neutrophil index) as a prognostic blood biomarker of mortality in early-stage sepsis patients [31]. Sepsis index using nCD64 and mHLA-DR was used in the determination of neonatal sepsis [12]. In our study, we found that nCD64-mHLADR index was significantly increased in sepsis patients.

Apart from phagocytic and oxidative burst activity of neutrophils, they also form NETs in response to the pathogens. This NETs formation occurs in two ways, i.e., lytic or non-lytic NETs. Classic late lytic NET gets induced by stimuli, such as PMA, antibodies causing activation of PKC or RAF–MEK–MAPK pathways, and PAD4 with ROS production, which enables citrullination of histones and subsequent chromatin decondensation. Neutrophils lose their viability after lytic NET formation. The early non-lytic NET gets induced by the stimuli recognition through TLR2, TLR4 or complement receptors independent of NADPH oxidase activation. Neutrophils remain alive and perform their functionality i.e., phagocytosis and chemotaxis even after the release of NETs [32,33]. Extracellular DNA acts as the backbone of NETs structure which gets detected by SYTOX green as it intercalates into cell-free DNA as well as enters dead cells, not live cells [34]. In our study, we have stimulated healthy neutrophils with patient’s plasma and LPS, PMA, and rIL-8 as positive controls and found that plasma from sepsis patients have more NETs formation as well as significantly increased cell-free DNA compared to *w*/*o* sepsis and HC plasma.

Sepsis patients showed an increase in a few of the pro-inflammatory cytokines i.e., IL-6, IL-8, IL-18, IL-33, IP-10, MIF, MIP1a, MIP1β, and MIP3a, along with the growth factors i.e., LEPTIN, GCSF, MCSF and ESelectin. Out of these increased cytokines and growth factors, IL-6, IL-8, IL-18, IP-10, MIF, and ESelectin were found decreased at day 3 and day 7, but IL-10, IL-33, MIP1a, MIP1β, MIP3a, and MCSF were increased compared to day 7. Whereas sepsis patients also showed a decrease in a few pro-inflammatory cytokines i.e., IL-1β, IL-2, IL-12p40, IL-17A, IL-27, TNF-α, and MCP-1; few anti-inflammatory cytokines i.e., IL-1RA, IL-4, IL-27, and TGF-β1; and growth factors, i.e., ENA78, Angiopoietin, TREM1, EOTAXIN, GM-CSF, HGF, PSELECTIN, MMP1, MMP8, MMP12, and MMP13. Some of the cytokines, i.e., IL-1β, TNF-α, and MCP-1 show higher levels in healthy controls which correlates well with the study of serum cytokine profiles in both healthy young and elderly population [35].

In addition to DNA, neutrophils also contain proteases and antimicrobial molecules such as histones and require various histone proteins for their activity towards pathogens, these include elastase, myeloperoxidase, cathepsin G, lactoferrin, pentraxin 3, gelatinase, proteinase 3, and others which have bactericidal activity [13]. In our study, we found many plasma proteins related to neutrophils activity, i.e., neutrophils degranulation, neutrophil activation involved in immune response, neutrophil-mediated immunity, and its granules. Upregulated proteins were DNAJC13, GPI, ANPEP, COTL1, GCA, GSTP1, PNP, PGAM1, TBMBS4X and APOA1, while downregulated proteins were AHSG, PROS1, SERPINA3, MPO, MMRN1 and DEFA1 in sepsis patients compared to *w*/*o* sepsis and HC. Out of these, a few of the upregulated proteins, i.e., GPI, TMSB4X, GCA, GSTP1, PNP, and COTL1 decreased at day 7 follow-up and all downregulated proteins increased at day 3 and day 7 compared to day 0.

Out of these upregulated and downregulated proteins, many proteins were known to play an important role in the innate immune system and neutrophils processes. ANPEP (Aminopeptidase *n* or CD13) is mainly present on the surface of neutrophils and gets upregulated by proinflammatory agonists, i.e., fMLF, IL-8, and TNF-α. The use of proinflammatory agonist TNF-α enhances the neutrophils apoptosis, which gets regulated by ANPEP in humans, and which can be reversed by the ANPEP inhibitors such as actinonin or bestatin [36,37]. The activated ANPEP also controls the LPS stimulated IL-8R on human neutrophils [38]. A novel EF-hand Ca^2+^-binding protein, i.e., GCA (Grancalcin) mainly expressed in neutrophils and monocytes or macrophages. GCA-deficient neutrophils cause the recruitment to the inflamed site and activation of neutrophils [39] and help in host defense against bacterial infections [40]. An increase in the expression of APOA1 (Apolipoprotein A–I) causes a decrease in the neutrophil functions, i.e., degranulation, oxidative burst, and superoxide production [41], whereas a decrease in APOA1 acts as a marker of liver fibrosis and improves survival in the sepsis mice model [42].

A deficient release of MPO in decompensated alcoholic cirrhosis causes a decrease in neutrophils antimicrobial ability, as ROS production by MPO helps in killing the microorganisms in chronic diseases [14,43,44]. In human neutrophils, a higher concentration of azurophilic granules with DEFA1 (α-defensins) is released through degranulation of activated neutrophils and directly kills phagocytosed microbes [45,46]. Down-regulation of serpina3 and serpina1 during mobilization causes a shift in the serine proteases and their inhibitors balance, which helps in the accumulation of active neutrophil serine proteases in the bone marrow that cleave and deactivate vital molecules for retention of hematopoietic progenitor cells [47].

Apart from innate immune response and neutrophil processes, many of these proteins are also involved in autophagy and apoptosis. During sepsis, autophagy uses a cellular adaptive defensive method for limiting cell damage and apoptosis, as well as to remove bacteria and pathogens. It plays an important role in neutrophils processes as neutrophil autophagic machinery gets activated by encountering various stimuli via phagocytosis-dependent or phagocytosis-independent signals. In addition, due to nutrient starvation in sepsis, neutrophil autophagy starts with the activation of the AMPK pathway [18,19]. Furthermore, autophagy and ROS generation are mainly required for chromatin decondensation in NETosis, and a fault in either pathway leads to apoptotic cell death instead of NETosis. Impaired autophagy causes a decrease in NETosis, leading to the survival of neutrophils, which then directly increases the mortality of sepsis patients [20].

DNAJC13 (DnaJ heat shock protein family (Hsp40) member C13) protein acts as a positive modulator of autophagy and maintains cellular homeostasis by ATG9A trafficking [48]. AHSG (Alpha-2-HS-glycoprotein) protein is mainly produced in the liver, and its level was decreased in sepsis patients. It stimulates the LC3 structure, as well as inhibits the LPS-induced increase of HMGB1 in autophagy and apoptosis manner acting as a negative regulator of HMGB1 in sepsis. AHSG also increases the phagocytosis of bacteria by neutrophils [49]. The role of TMSB4X (Thymosin beta 4) was already known in inflammation and liver diseases. It mainly enhances the autophagy process via HIF-1α and activates HSC during liver fibrosis via PI3K/AKT signaling pathway. An increase in the TMSB4X levels acts as a prognosis marker of NASH and ACLF, which is also negatively related to the oxidation state of the liver. In sepsis, a significant decrease in TMSB4X levels activates the inflammatory cascade and is correlated with a decrease in mortality [50,51]. SERPINA3 (α1-antichymotrypsin) mediated apoptosis, autophagy, and aerobic glycolysis were regulated by circSERPINA3 by competitively binding to the miR-653-5p and finally recruiting BUD13 [52]. PROS1 (Vitamin K-dependent protein) protein plays a role via induction of both autophagy and apoptosis via promoting the PI3K/AKT/HIF-1α mediated glycolysis that causes the AMPK dependent autophagic cell death and apoptosis via FASL, caspase8, BAD and survivin [53], while PGAM1 (Phosphoglycerate mutase 1) protein plays a role in the glycolysis process but acts as a downstream target of PI3K/AKT/mTOR/HIF-1α pathway as mTOR-PGAM1 signaling contributes to the development of Warburg effect (aerobic glycolysis) via HIF-1α [54]. Further, validating DNAJC13, AHSG, and TMSB4X proteins at the RNA level have provided the strength to this study. Comparing the age and other parameters of the study groups were varying; therefore, median with range given which acts as a limitation in this study.

## 5. Conclusions

We conclude that the faulty functionality of neutrophils may be due to the autophagy proteins, i.e., DNAJC13, AHSG, TMSB4X, PROS1, and SERPINA3, which can be used as therapeutic targets in decompensated cirrhosis patients with sepsis.

## Figures and Tables

**Figure 1 cells-11-01745-f001:**
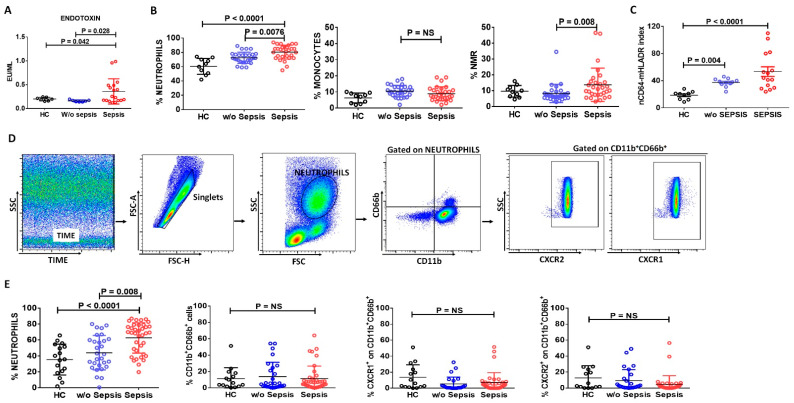
**Expression of neutrophils, monocytes, and nCD64-mHLADR Index in patient groups.** Scatter dot plot shows (**A**) Endotoxin levels, (**B**) differential leukocyte count (DLC): neutrophils, monocytes, and neutrophils to monocyte ratio (NMR) by analyzing complete blood count and (**C**) nCD64-mHLADR Index between the groups. (**D**) Sequential gating strategy for identification of neutrophils and their activated markers, i.e., CXCR1, CXCR2 and CD66b using flow cytometry. (**E**) Scatter dot plot shows the percent of neutrophils, CD11b^+^CD66b^+^, percent of CXCR1^+^ and CXCR2^+^ on CD11b^+^CD66b^+^ between the groups using flow cytometry. Results are expressed as the mean ± SD; One-way ANOVA/Kruskal–wallis test followed by probability adjustment by the Mann–Whitney.

**Figure 2 cells-11-01745-f002:**
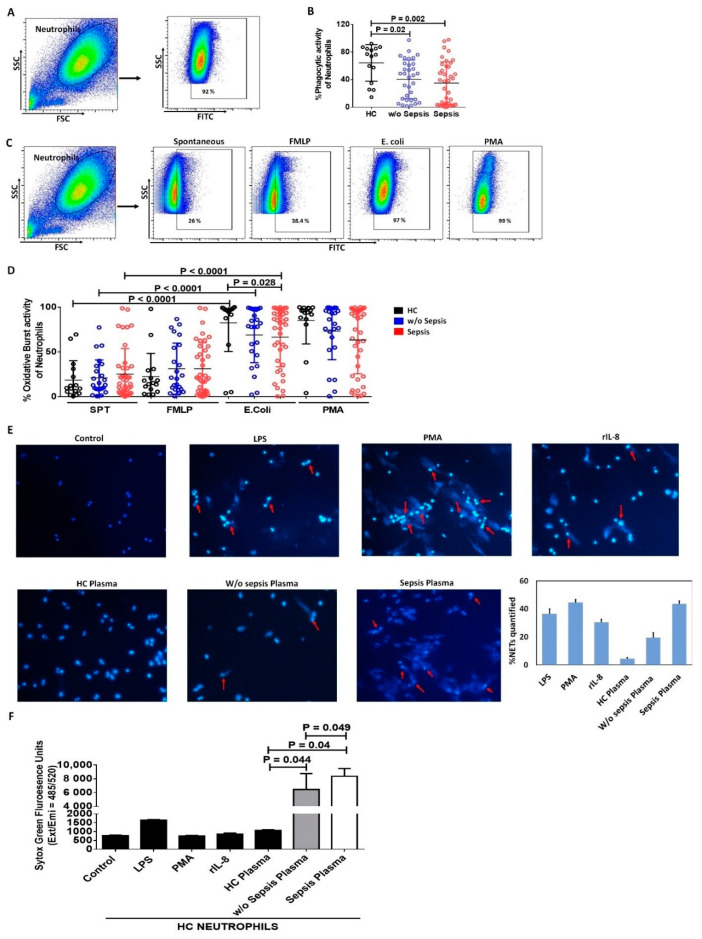
**Functionality of neutrophils.** (**A**) Gating strategy for phagocytic activity of neutrophils. (**B**) Scatter dot plot shows phagocytic activity of neutrophils between the groups. (**C**) Gating strategy for oxidative burst activity of neutrophils both spontaneously and stimulation with FMLP, *E. coli* and PMA. (**D**) Scatter dot plot shows oxidative burst activity of neutrophils between the groups. (**E**) Representative images of NETs in healthy control neutrophils without and with stimulation i.e., LPS, PMA, rIL-8, HC plasma, *w*/*o* sepsis and sepsis patient plasma at 40× along with the % NETs quantification using ImageJ software. (**F**) Cell free DNA was estimated using sytox green fluorescence units captured on exi/emi *=* 485/520 in supernatant without and with stimulation. SPT (spontaneous), LPS (Lipopolysaccharide), PMA (Phorbol 12-myristate 13-acetate), rIL-8 (recombinant IL-8). Results are expressed as the mean ± SD; One-way ANOVA/Kruskal–wallis test followed by probability adjustment by the Mann–Whitney.

**Figure 3 cells-11-01745-f003:**
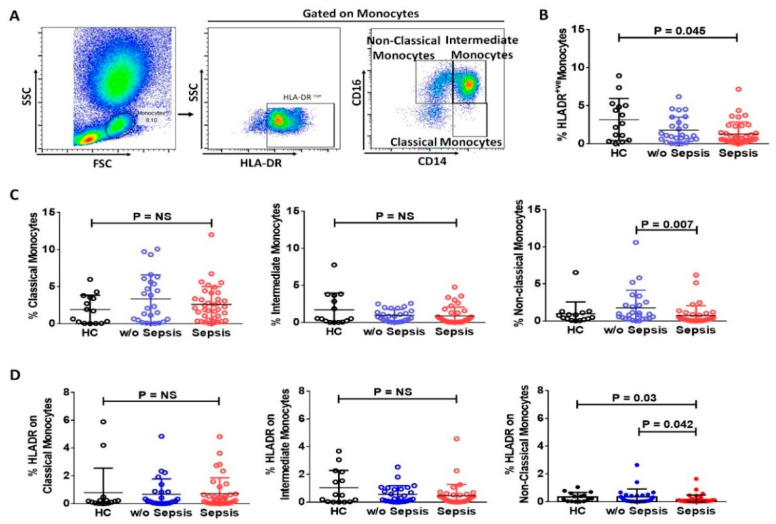
**Expression of monocytes and its sub-types in patient groups.** (**A**) Sequential gating strategy for identification of monocytes and their sub-types using flow cytometry. Scatter dot plot shows the expression of (**B**) HLA-DR^+ve^ on monocytes, (**C**) Sub-types i.e., classical, intermediate, and non-classical monocytes. (**D**) HLA-DR expression on classical, intermediate, and non-classical monocytes. Results are expressed as the mean ± SD; One-way ANOVA/Kruskal–wallis test followed by probability adjustment by the Mann–Whitney.

**Figure 4 cells-11-01745-f004:**
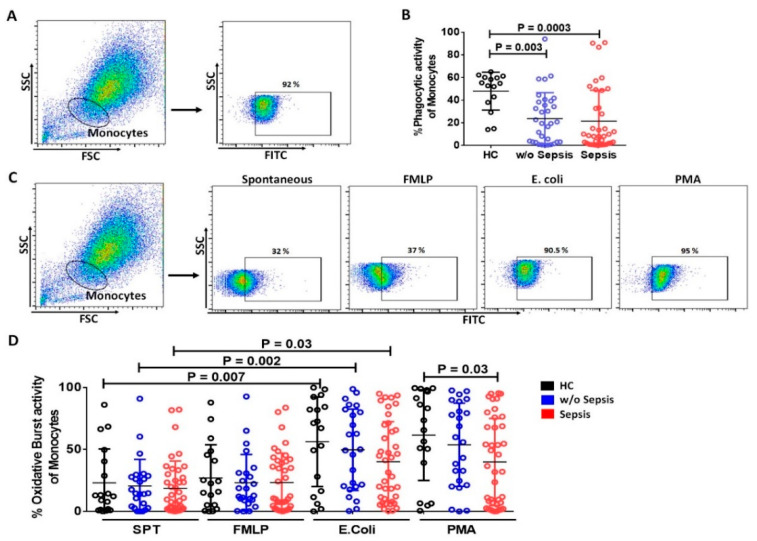
**Functionality of monocytes.** (**A**) Gating strategy for phagocytic activity of Monocytes. (**B**) Scatter dot plot shows phagocytic activity of monocytes between the groups. (**C**) Gating strategy for oxidative burst activity of monocytes spontaneously and stimulation with FMLP, *E. coli* and PMA. (**D**) Scatter dot plot shows oxidative burst activity of monocytes between the groups. SPT (spontaneous). Results are expressed as the mean ± SD; One-way ANOVA/ Kruskal–wallis test followed by probability adjustment by the Mann–Whitney.

**Figure 5 cells-11-01745-f005:**
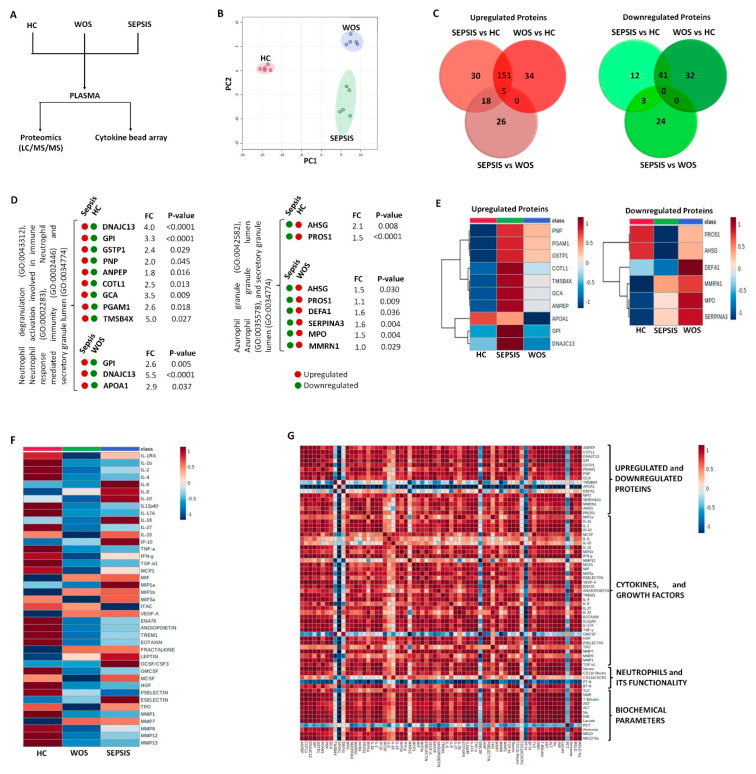
**Identification of proteins, and cytokines linked with neutrophils.** (**A**) Schematic representation of the work pipeline followed for proteomics and cytokine bead array. (**B**) Principal component analysis (PCA) between the groups. (**C**) Venn diagram showing total upregulated and downregulated proteins between the groups. (**D**) Few upregulated and downregulated proteins related to neutrophil activity which were found significant between the groups. (**E**) Heatmap showing these upregulated and downregulated proteins between the groups. (**F**) Cytokine bead array analysis between the groups. (**G**) Correlation matrix among %neutrophils, proteins, cytokines, and biochemical parameters in sepsis patients. It was obtained by deriving a Pearson correlation coefficient. Red is a positive correlation, and blue is a negative correlation.

**Figure 6 cells-11-01745-f006:**
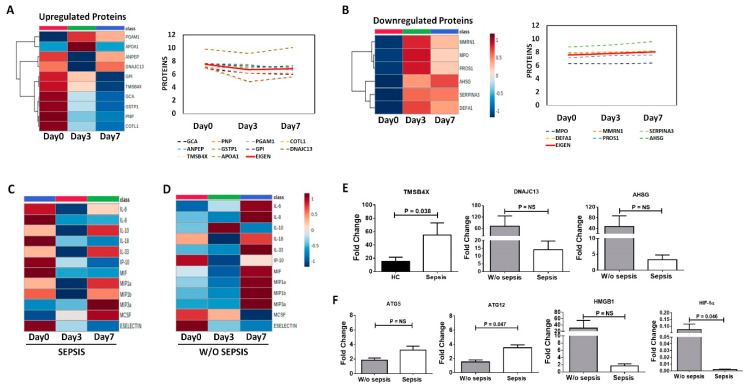
**Proteins and cytokines at different time points.** Heatmap and Line diagram showing the expression of (**A**) Upregulated and (**B**) downregulated proteins in sepsis patients compared at Day 0, Day 3 and Day 7. The line diagram shows the log values of all the proteins (dotted line) and Eigen value of all the proteins (solid line). An eigen values are unit-less and are derived via PCA. Expression of cytokines, chemokines, and growth factors in (**C**) sepsis and (**D**) *w*/*o* sepsis at different time points i.e., Day 0, Day 3 and Day 7. Column bar graphs show fold change of (**E**) TMSB4X, DNAJC13, and AHSG, and (**F**) ATG5, ATG12, HMGB1 and HIF-1α using QRT-PCR.

**Table 1 cells-11-01745-t001:** Baseline clinical, as well as biochemical characteristics of study groups.

MEDIAN and RANGE	Healthy Control (HC, *n =* 15)	DC *w*/*o* Sepsis (*n =* 30)	DC with Sepsis (*n =* 40)	*p* Value–*w*/*o* and with Sepsis
Age	32 (20–40)	47 (22–60)	43 (29–60)	0.02
Male: Female	11:4	25:5	38:2	-
Total Bilirubin (mg/dL)	1 (0.3–1.5)	6.8 (1.6–22)	17.9 (2.2–31.7)	<0.00
AST (IU/mL)	20 (5–40)	62 (31–510)	118 (35–1037)	0.04
ALT (IU/mL)	25 (10–40)	37 (20–634)	46 (11–233)	NS
INR (s)	1 (0.8–1.2)	1.9 (1.1–3.3)	2.7 (1.3–6.75)	<0.00
PCT (ng/mL)	0.8 (0.2–2)	0.4 (0.04–3.25)	3 (0.07–88)	0.04
MELD Na	8 (6–10)	26 (10–37)	34 (29–40)	<0.00
Lactate (mmol/L)	1.5 (1–2)	1.4 (0.6–5.2)	2.1 (0.2–13.3)	0.02
Sodium (mmol/L)	140 (136–145)	133 (124.3–142.7)	129 (119–150)	NS
Creatinine (mg/mL)	0.6 (0.2–1)	0.86 (0.3–2.9)	1.57 (0.3–4.78)	0.03
SOFA score	-	7 (4–11)	12 (10–16)	<0.00
SIRS Criteria				
TLC (10^9^ L)	6 (4–11)	6.6 (3.1–19.8)	14 (2.7–43.6)	0.01
PULSE (/minute)	70 (60–100)	76 (60–110)	94 (62–132)	0.01
RR (/minute)	14 (12–16)	20 (16–24)	22 (14–34)	NS
TEMPERATURE	98 (97–99)	98.4 (97–98.9)	98.4 (96–100)	NS

DC (Decompensated cirrhosis), AST (aspartate aminotransferase), ALT (Alanine transaminase), INR (The international normalized ratio), PCT (procalcitonin), MELD Na (Model for End-stage Liver Disease Sodium), SOFA (Sequential organ failure assessment), SIRS (systemic inflammatory response syndrome), TLC (Total leukocyte count), RR (Respiratory rate).

## Data Availability

All immune and protein data generated during the study are available from the corresponding author upon request.

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
