# Peer review of "Plasma Proteomic Analysis Identified Proteins Associated with Faulty Neutrophils Functionality in Decompensated Cirrhosis Patients with Sepsis"

_cells, 2022, doi:10.3390/cells11111745_

Round 1
Reviewer 1 Report
The manuscript titled, “Plasma proteomic analysis identified proteins associated with 2 faulty neutrophils functionality in decompensated cirrhosis patients with sepsis” by Sehgal, et al., identified differentially expressed/regulated proteins in the plasma of healthy controls , patients with decompensated cirrhosis and patients with decompensated cirrhosis + sepsis, and discovered targets which were uniquely expressed in patients with sepsis as compared to other groups and propose therapeutic targeting of the discovered proteins in sepsis patients. The findings from this study is interesting and tries to address important questions in the field.
Major Comments:
- Manuscript must go extensive revision for English.
- Results derived from the different timepoints of study are interesting (Day 0, 3 and 7) but are not further addressed in the discussion section. Including endotoxin levels in plasma would be useful to understand neutrophil and monocyte dynamics. Moreover, data analysys based on sepsis treatments might shed light on decreased monocyte and neutrophil populations at day 7.
- The work presented is primarily descriptive with limited novelty in the field. Authors should reformulate the aim of the study “the aim to evaluate proteins related to neutrophils functionality in sepsis” in a catchier question to address.
- NET staining images (Figure 1E) should be improved. Higher magnification images or alternative markers such as MPO or Histone H3 might also provide NETosis quantification.
- Authors claim that autophagy regulates neutrophil functionality based on autophagy-related proteins found in plasma. Further experiments should be performed to validate this association.
- Reviewer is concerned about the relation between proteins found in the systemic circulation with neutrophil activity. Even though some proteins might be specifically secreted by neutrophils, validation of key finding in supernatants from isolated neutrophils will add extra value to the work.
- Fig1 D. The gating strategy used by the authors seems unconventional. For example, after gating out dead cells and selecting singlets, further gating on CD11b+ CD66b+ cells to select neutrophils and then looking for CXCR1/CXCR2 will be one of the traditional approaches. While it looks clear that the cells were chosen based on their high side scattering property, it’s unclear about the overall gating strategy used. If the intention is to show CXCR1/CXCR2 levels on neutrophils, then this strategy has to be revisited.
- Line 389 “Validation of neutrophil functionality in circulation” was this intended to be a part of line 390? If this sentence is to be used, the word validation has to replaced, as there is no validation of any targets identified from the proteomics screen.
- Line 394-397, the use of “differentially expressed” word is rather confusing in this context. For instance, in Figure 5.C, if 30 genes are present in sepsis Vs HC, rephrasing the sentence to “30 uniquely expressed in sepsis Vs HC” will help understand better, if this is what the authors intend to explain.
- F it’s interesting to note that in the heat map presented, expression of certain cytokines go up (IL6, IL8, IL18, MIF, MIP1a, MIP1b, etc), while the levels of IL-1b, TNFa, seem to be decreasing. Also, it’s rather surprising to note such high levels of IL-1b, TNFa and MCP-1 in healthy controls. This is not discussed and needs to be clarified.
- That authors conclude with the data from the proteomic analysis showing increase in the levels of DNAJC13, AHSG, TMSB4X, PROS1 and SERPINA3, which potentially contribute to autophagy and NETs formation. This is an interesting evidence but is there any evidence that targeting one of these factors reducing autophagy/NETOsis? In Line 586, the authors mention lack of validation as a limitation of the study, it will be in great support of the presented data if even one target is validated.
Minor comments:
- Line 534- Please check for sentence reconstruction, “ This TNFa….”
- Line 560- Please check for sentence reconstruction, “……neutrophil autophagy begun via AMPK pathway”
- Missing reference for statement in page 2 line 69.
- Full name for abbreviations should be stated the first time abbreviation appears in text e.g. SIRS in page 2 line 89.
- Subsections in the methodology part should be reorganized in a logical manner, expression of neutrophilic CD64 or oxidative burst activity is described before neutrophil isolation. Temperature at which blood was collected, maintained and neutrophils were isolated is missing.
- What is the difference between panel 1A and 1E, please state clearly in the figure legend and MM sections.
- Limitation of the study are not experiments that have not been performed but those flaws or obstacles from the methodology/analysis from the current data in the manuscript.
- It would be interesting to perform sub analysis based on cirrhosis etiology, decompensating events (besides sepsis) or treatments in the sepsis group to further understant the phenotype of neutrophils in this specific populations.
Author Response
REVIEWER 1
The manuscript titled, “Plasma proteomic analysis identified proteins associated with 2 faulty neutrophils functionality in decompensated cirrhosis patients with sepsis” by Sehgal, et al., identified differentially expressed/regulated proteins in the plasma of healthy controls , patients with decompensated cirrhosis and patients with decompensated cirrhosis + sepsis, and discovered targets which were uniquely expressed in patients with sepsis as compared to other groups and propose therapeutic targeting of the discovered proteins in sepsis patients. The findings from this study is interesting and tries to address important questions in the field.
Major Comments:
1. Manuscript must go extensive revision for English.
Answer: Thanks for the suggestions. We have made corrections throughout the manuscript.
2. Results derived from the different timepoints of study are interesting (Day 0, 3 and 7) but are not further addressed in the discussion section. Including endotoxin levels in plasma would be useful to understand neutrophil and monocyte dynamics. Moreover, data analysys based on sepsis treatments might shed light on decreased monocyte and neutrophil populations at day 7.
Answer: Thanks for the correcting us. We indeed added endotoxin levels in results section (line 291) and discussed its correlation with neutrophils and monocytes in the discussion section (line 580). With the addition of new references, sequence of references is rearranged in the revised manuscript.
We have also discussed different time points (day 0, 3 and 7) data in the discussion section (line 585, 613 and 634) in the revised manuscript.
As treatment of sepsis patients were same with broad spectrum antibiotics, some patients resolve the infection/sepsis while others were not able to resolve causing mortality in the patients. As per the reviewer suggestions, we have compared the neutrophils and monocytes population in patients who have resolved or not resolved the infection. Results are given in supplementary Figure 3 and 7 which is discussed in discussion section (line 587) in the revised manuscript.
3. The work presented is primarily descriptive with limited novelty in the field. Authors should reformulate the aim of the study “the aim to evaluate proteins related to neutrophils functionality in sepsis” in a catchier question to address.
Answer: We appreciate reviewer concern, and we have refined the aim as per reviewer suggestion in the revised manuscript.
4. NET staining images (Figure 1E) should be improved. Higher magnification images or alternative markers such as MPO or Histone H3 might also provide NETosis quantification.
Answer: Thanks for the suggestions. NET staining images are in Figure 2E and as per the reviewer suggestions, we have replaced the NETs staining images with higher magnification images (40x) in the revised manuscript.
In NETosis, extracellular DNA gets expelled from neutrophils, which was detected by using sytox green in the supernatants as cell-free DNA indicating one of the methods to quantify NETosis.
Further upon reviewer suggestion, we have analyzed NETosis quantification using semi-quantitative method with ImageJ software. Analyzed Figure is added in Figure 2E in the revised manuscript.
5. Authors claim that autophagy regulates neutrophil functionality based on autophagy-related proteins found in plasma. Further experiments should be performed to validate this association.
Answer: Thanks for the suggestions. Taking reviewer suggestions into consideration, we have validated autophagy related proteins i.e DNAJC13, TMSB4X, and AHSG along with ATG5, ATG12, HMGB1 and HIF-1α using RT-PCR in the stored neutrophils. The results are discussed in Figure 6E-F in the revised manuscript.
6. Reviewer is concerned about the relation between proteins found in the systemic circulation with neutrophil activity. Even though some proteins might be specifically secreted by neutrophils, validation of key finding in supernatants from isolated neutrophils will add extra value to the work.
Answer: We agree with the reviewer concerns regarding the validations required in this manuscript. Validation of proteins in the supernatants from isolated neutrophils was not possible as neutrophils have a short life span and require fresh isolated neutrophils for the validations.
However, we have validated few of the key plasma proteins i.e DNAJC13, TMSB4X, and AHSG using RT-PCR in the stored neutrophils. The results are discussed in Figure 6E in the revised manuscript.
7. Fig1 D. The gating strategy used by the authors seems unconventional. For example, after gating out dead cells and selecting singlets, further gating on CD11b+ CD66b+ cells to select neutrophils and then looking for CXCR1/CXCR2 will be one of the traditional approaches. While it looks clear that the cells were chosen based on their high side scattering property, it’s unclear about the overall gating strategy used. If the intention is to show CXCR1/CXCR2 levels on neutrophils, then this strategy has to be revisited.
Answer: We appreciate reviewer’s concern. We have addressed the point and revisited the gating strategy accordingly i.e after selecting singlets, we have gated on CD11b+CD66b+ cells and further showed expression of CXCR1 and CXCR2 on CD11b+CD66b+ cells. Gating strategy and study data given in Figure1 and discussed in the result section in the revised manuscript.
8. Line 389 “Validation of neutrophil functionality in circulation” was this intended to be a part of line 390? If this sentence is to be used, the word validation has to replaced, as there is no validation of any targets identified from the proteomics screen.
Answer: Thanks for correcting us, we agree with the reviewers that there is no validation of neutrophils functionality, so we have removed the sentence in line 389 “validation of neutrophil functionality in circulation”.
9. Line 394-397, the use of “differentially expressed” word is rather confusing in this context. For instance, in Figure 5.C, if 30 genes are present in sepsis Vs HC, rephrasing the sentence to “30 uniquely expressed in sepsis Vs HC” will help understand better, if this is what the authors intend to explain.
Answer: Thanks for correcting us. As per the reviewer suggestions, we have rephrased the sentence to “30 upregulated proteins were uniquely expressed in sepsis and HC” and “12 downregulated proteins were uniquely expressed in sepsis and HC”. Corrections were made accordingly in the revised manuscript.
10. F it’s interesting to note that in the heat map presented, expression of certain cytokines go up (IL6, IL8, IL18, MIF, MIP1a, MIP1b, etc), while the levels of IL-1b, TNFa, seem to be decreasing. Also, it’s rather surprising to note such high levels of IL-1b, TNFa and MCP-1 in healthy controls. This is not discussed and needs to be clarified.
Answer: Thanks for the suggestions. We agree with the reviewers that we have only discussed increased cytokines in sepsis, not the decreased cytokines. Now we have corrected it and discussed in the result as well as discussion (line 613) in the revised manuscript.
As per the reviewer concerns regarding high level of some of cytokines i.e IL-1β, TNF-α and MCP-1 in healthy controls, we have used human custom PROCARTAPLEX (40 plex, Thermofischer #PPX-40-MXNKR22) whose lowest detection limit already given in Supplementary Table 1. We want to clarify that range of IL-1β, and MCP-1 levels in our healthy control individuals were comparative to the study of cytokines in healthy individuals [Kim et al. Journal of Translational Medicine 2011]. Details are discussed (line 621) in the revised manuscript.
11. That authors conclude with the data from the proteomic analysis showing increase in the levels of DNAJC13, AHSG, TMSB4X, PROS1 and SERPINA3, which potentially contribute to autophagy and NETs formation. This is an interesting evidence but is there any evidence that targeting one of these factors reducing autophagy/NETOsis? In Line 586, the authors mention lack of validation as a limitation of the study, it will be in great support of the presented data if even one target is validated.
Answer: We agree with the reviewer concerns. As per the reviewer suggestions, we have validated few of the key plasma proteins i.e DNAJC13, TMSB4X, and AHSG using RT-PCR in the stored neutrophils. The results are discussed in Figure 6E in the revised manuscript.
Minor comments:
1. Line 534- Please check for sentence reconstruction, “ This TNFa….”
Answer: Thanks for correcting us. Corrections made as per the suggestions.
2. Line 560- Please check for sentence reconstruction, “……neutrophil autophagy begun via AMPK pathway”.
Answer: Thanks for correcting us. Corrections made as per the suggestions.
3. Missing reference for statement in page 2 line 69.
Answer: Thanks for your concern but reference number 18-20 covers reference for the whole paragraph (line 66-73).
4. Full name for abbreviations should be stated the first time abbreviation appears in text e.g. SIRS in page 2 line 89.
Answer: Thanks for correcting us. Abbreviation of SIRS added in page 2 (line 89) in the revised manuscript.
5. Subsections in the methodology part should be reorganized in a logical manner, expression of neutrophilic CD64 or oxidative burst activity is described before neutrophil isolation. Temperature at which blood was collected, maintained and neutrophils were isolated is missing.
Answer: Thanks for your concern, we here want to clarify that the methodology was organized in step wise manner as expression of neutrophilic CD64 and monocytic HLA-DR [BD Biosciences #340768 and #340827] was checked in part of whole blood. Further, phagocytic and oxidative burst activity [Celonic, Switzerland #341060 and #341058] were also checked in whole blood samples before we have isolated the neutrophils from the rest portion of whole blood.
After using whole blood for the above-mentioned experiments, rest blood was used for neutrophils isolation.
Later, Netosis experiments were done using isolated neutrophils.
Peripheral blood was collected at room temperature from subjects and immediately processed for all experiments. As neutrophils has a lifespan of only 5.4 days [Hidalgo et al., Trends Immunol. 2019 and Pillay et al., Blood. 2010], neutrophil isolation and functionality was measured immediately and rest all the experiments were performed accordingly.
Details were added in blood sampling and neutrophil isolation section in material and methods as per the reviewer suggestions.
6. What is the difference between panel 1A and 1E, please state clearly in the figure legend and MM sections.
Answer: Thanks for your concern. As we have added endotoxin levels in Figure 1 as Panel A, now Panel B shows the differential leukocyte count (DLC): neutrophils, monocytes, and neutrophils to monocyte ratio (NMR) by analyzing complete blood count using clinical hematology instrument.
However, further whole blood was subjected to flow cytometry analysis for percent total neutrophils, CD11b+CD66b+ neutrophils and expression of CXCR1 and CXCR2 on CD11b+CD66b+ neutrophils in whole blood.
Revised figure and results are now clearly mentioned the revised manuscript.
7. Limitation of the study are not experiments that have not been performed but those flaws or obstacles from the methodology/analysis from the current data in the manuscript.
Answer: Thanks for your concern. As now we have validated few of proteins at RNA levels also, therefore we have removed limitations of this study.
8. It would be interesting to perform sub analysis based on cirrhosis etiology, decompensating events (besides sepsis) or treatments in the sepsis group to further understand the phenotype of neutrophils in this specific populations.
Answer: Thanks for your concern.
As mentioned in the result section (line 288), alcohol was the predominant etiology (70%) in our cirrhosis patients’ groups and rest etiology with smaller number of patients involves NASH and cryptogenic.
As per the reviewer suggestions, we have done sub-analysis of neutrophils, CD11b+CD66b+ neutrophils and expression of CXCR1 and CXCR2 on CD11b+CD66b+ neutrophils in the patients based on alcohol and other etiologies (NASH and cryptogenic). Results are given in supplementary figure 2A and discussed in the revised manuscript.
Further, as per the reviewer suggestions regarding decompensating event besides sepsis, we have done sub-analysis of neutrophils, CD11b+CD66b+ neutrophils and expression of CXCR1 and CXCR2 on CD11b+CD66b+ neutrophils in the patients based on AKI as a decompensating event (besides sepsis) in decompensated cirrhosis patients with sepsis. Results are given in supplementary figure 2B and discussed in the revised manuscript.
Reviewer 2 Report
The control group has an average age of 32 years, the cirrhosis groups are on average 10 years older. This can be a hindrance for studies on the immune system since the immune system naturally changes through senescence. A control group with the same average age would have been better.
SIRS criteria should no longer be used to define sepsis, as they are non-specific and are reserved as a separate entity for non-infectious systemic inflammatory reactions (trauma, pancreatitis). It would be better to use the SOFA criteria.
Author Response
- The control group has an average age of 32 years, the cirrhosis groups are on average 10 years older. This can be a hindrance for studies on the immune system since the immune system naturally changes through senescence. A control group with the same average age would have been better.
Answer: We appreciate reviewer concern regarding the age of our study subjects. As the age and other parameters of the study groups was varying, therefore instead of mean values we have given median with range.
Further for sepsis patient group, without sepsis patients act as the clinical control which shows no difference in age compared to sepsis group.
- SIRS criteria should no longer be used to define sepsis, as they are non-specific and are reserved as a separate entity for non-infectious systemic inflammatory reactions (trauma, pancreatitis). It would be better to use the SOFA criteria.
Answer: Thanks for correcting us. We here want to clarify that the SIRS criteria solely was not used for the characterization of sepsis in cirrhosis patients. As per the reviewer suggestions we have calculated SOFA score for the study groups and given in table 1.
Round 2
Reviewer 2 Report
The authors have included the SOFA score in their analysis and thus eliminated a major point of criticism. Unfortunately, the criticism of age and liver cirrhosis compared with the control group cannot be changed retrospectively, but it should be listed as a limitation. The authors attempt an explanation by postulating that the sepsis group and the control group are of equal age. Therefore, the manuscript can be considered for publication.
Author Response
The authors have included the SOFA score in their analysis and thus eliminated a major point of criticism. Unfortunately, the criticism of age and liver cirrhosis compared with the control group cannot be changed retrospectively, but it should be listed as a limitation. The authors attempt an explanation by postulating that the sepsis group and the control group are of equal age. Therefore, the manuscript can be considered for publication.
Answer: Thanks for correcting us. We have given age and other parameters between HC and study groups as limitation in the discussion section of the revised manuscript.